# The Prediction of the Performance of a Twisted Rudder

**Ilryong Park** [1,*], **Bugeun Paik** [2], **Jongwoo Ahn** [2] **and Jein Kim** [1]

[1] Department of Naval Architecture and Ocean Engineering, Dong-Eui University, Busan 47340, Korea; jein@deu.ac.kr

[2] Korea Research Institute of Ships & Ocean Engineering, Daejeon 34103, Korea; ppaik@kriso.re.kr (B.P.); ajwprop@kriso.re.kr (J.A.)

\* Correspondence: irpark@deu.ac.kr; Tel.: +82-051-890-2595

**Abstract:** A new design approach using the concept of a twisted rudder to improve rudder performances has been proposed in the current paper. A correction step was introduced to obtain the accurate inflow angles induced by the propeller. Three twisted rudders were designed with different twist angle distributions and were tested both numerically and experimentally to estimate their hydrodynamic characteristics at a relatively high ship speed. The improvement in the twisted rudders compared to a reference flat rudder was assessed in terms of total cavitation amount, drag and lift forces, and moment for each twin rudder. The total amount of surface cavitation on the final optimized twin twisted rudder at a reference design rudder angle decreased by 43% and 34.4% in the experiment and numerical prediction, respectively. The total drag force slightly increased at zero rudder angle than that for the twin flat rudder but decreased at rudder angles higher than 4° and 6° in the experiment and numerical simulation, respectively. In the experimental measurements, the final designed twin twisted rudder gained a 5.5% increase in the total lift force and a 37% decrease in the maximum rudder moment. Regarding these two performances, the numerical results corresponded to an increase of 3% and a decrease of 66.5%, respectively. In final, the present numerical and experimental results of the estimation of the twisted rudder performances showed a good agreement with each other.

**Keywords:** twisted rudder; rudder performance; cavitation performance; CFD; large cavitation tunnel





## 1. Introduction

Rudders operate in the complex interactive flow between hull, propeller, and rudder, which determines the maneuverability, self-propulsion, and cavitation performances of ships along with hull forms and propellers. The cavitation performance of rudders can be improved by reducing the influence of the crossflow induced by the propeller. This is accompanied by a possible increase in the lift-drag ratio, which helps to improve the ship's maneuvering performance. The propulsion efficiency can be improved, to some extent, by recovering the rotational energy between the propeller and rudder.

In recent, energy efficiency has become an important issue in ship design and operation. In general, hull form and propeller blade optimizations have been considered to improve resistance and propulsion efficiencies [1,2]. For the same purposes, various types of energy-saving devices (ESDs) have been developed focused on optimized hydrodynamic interaction between hull form, propeller, and rudder. The improvement of ship hydrodynamic performance through ESDs can be found in various cases ranging from resistance, propulsion, cavitation, and maneuvering performances [2–7]. According to the study by Carlton [2], zones for ESD implementations to recover the energy losses from propellers can be classified as pre- [8–10], in- [11–13], and post-propeller plane [14,15]. Meanwhile, Mewis and Deichmann [8] and Watson [14] also classified two different types of approaches adoptable for ESDs: pre-rotating and post propeller recovery. In general, the estimation of the hydrodynamic performance of developed ESDs has been done through computational

fluid dynamics (CFD) simulations and/or model tests performed in a towing tank and a cavitation tunnel. It is reported that discrepancies have been found between CFD and ITTC 1978 prediction concerning the scale effects on ESDs' performances estimated at model scale [16], and the need for CFD simulations at full-scale conditions and sea trials to assess these scale effects has been discussed [17]. There is limited sea trial data in the public domain, but it is clear that several ESDs have been fitted to operational ships and attempted in full-scale trials [18].

Shen et al. [18] investigated the scale effects on ships with ESDs using CFD simulations and reported the reliability of CFD data at full scale compared to model scale. Mori et al. [19], Ohtagaki et al. [20], and Okamoto et al. [21] studied a rudder bulb, or an additional device in a fin on the rudder bulb to retrieve the energy loss behind the propeller. These studies showed that rudder bulbs devised increase propulsion efficiency by reducing hub drag, which is caused by decreased separation and pressure pulse. Liu et al. [22] showed the energy-saving effect of the combination of rudder bulb and rudder thrust fin is better than that of rudder bulb at full-scale trials. Kanemaru et al. [23] proposed two kinds of newly developed rudders that obtain the low drag effectively comparing with the conventional rudder. Chen et al. [24] employed a parametric geometry design using the non-uniform rational B-splines (NURBS) technique to obtain the best rudder geometry for improving propulsive efficiency, where rudder optimization was performed based on CFD. Rhee et al. [25] developed rudder gap flow-blocking devices to suppress rudder gap cavitation and carried out cavitation observation and pressure measurement in a cavitation tunnel for the estimation of their cavitation performances. Paik et al. [26] investigated the unsteady cavity patterns around the gap of the conventional and newly developed semi-spade rudders for marine ships using a high-speed CCD camera, time-resolved particle image velocimetry (PIV) analysis, and pressure measurements. The relationship between the cavitation phenomenon on the rudder surface and the rudder angles was analyzed by Paik et al. [27], where PIV technology was adopted to observe the flow field between propeller and rudder. Reichel [28] carried out an experimental investigation of the effect of rudder location on the propulsion efficiency of a single-screw, single-rudder container ship. Krasilnikov et al. [29] experimentally analyzed the hydrodynamic characteristics of the propeller-rudder system at low-speed operation.

The problem of rudder cavitation has been an issue due to the type of rudder that has been adopted to most large ships [30]. Thus, full-spade rudders have been used for high-speed vessels and researched to avoid cavitation problems. Nishiyama [31] provided an empirical formula of inflow angle for twisted rudder generation and discussed the effectiveness of reaction rudder on rudder cavitation. Shen et al. [32] proposed a rudder design method to improve rudder cavitation performance and tested a twisted rudder in a large cavitation channel concerning rudder surface cavitation and cavitation inception speed. Kim et al. [33] and Choi et al. [34] developed twisted full-spade rudders based on incoming flow angles for a large container ship to recover the swirl energy for the propeller slipstream. Kim et al. [35] designed three twisted rudders for a large container carrier and verified the speed performances of the ship improved by those twisted rudders through model tests in a towing. Furthermore, they analyzed the change in self-propulsion factors by the twisted rudders. Ahn et al. [36] developed a twisted rudder to overcome the cavitation problems of large container carriers in the way of the leading edge also considering manufacturing productivity. They called this rudder the X-Twisted rudder and verified its hydrodynamic performances through various model tests such as resistance, self-propulsion, cavitation, and maneuvering tests. Sun et al. [37] performed numerical simulations and cavitation tunnel experiments to predict the hydrodynamic performance of twisted rudders and verify the energy-saving effect. Kim et al. [38] designed a twisted rudder by using the genetic algorithm and investigated cavitation behaviors on the rudder surface. A wavy twisted rudder superior to a conventional full-spade twisted rudder with regard to lift-drag ratio and stall delay was proposed and verified by Shin et al. [39].

In general, most research on twisted rudders has been carried out for single-screw ships. However, this study focused on a twin twisted rudder of a surface combatant and investigated its effects on hydrodynamic performances. In the current work, a novel design of a twisted rudder has been proposed to improve the rudder performance of an existing surface combatant. Our approach involved prediction and correction of twist angle distribution which compensate for the effect of inflow angles induced by the propeller. While developing the design of the twisted rudder, our main focus was to improve the rudder cavitation performance without any loss of lift performance. The primary design criteria are a decrease in rudder moment and a decrease in the drag force of the twisted rudder up to a certain level which is not harmful to the ship's self-propulsion performance. Three twisted rudders with three different twist angle distributions were designed using the results of the CFD simulations and tested both numerically and experimentally to estimate their hydrodynamic performances at a relatively high ship speed. Using the same test setup and procedures, a reference flat rudder was also tested to validate the effect of twist angles on the rudder performance. The numerical and experimental evaluations mainly focused on the rudder surface cavitation, rudder forces, and moment. The initial twist angle distribution was predicted from numerical simulation and its accuracy was estimated. A method of correction of the predicted initial inflow angles induced by the propeller was proposed and applied to obtain the final corrected distribution of twist angles. An intermediate distribution of the twist angles between the initial and final corrected twist angles was produced to see the variation effect of the twist angle on rudder hydrodynamic performances. In the current paper, the improvement of the twisted rudders compared with the reference flat rudder is discussed in terms of total cavitation amount, drag, lift, and moment for each twin rudder.

## 2. Experimental Approaches

Model tests to analyze the hydrodynamic performance of twisted rudders including a reference flat rudder were carried out in the large cavitation tunnel (LCT) at the KRISO. The main dimensions of the LCT in the length (L), width (W), and height (H) directions are 60 m × 6.5 m × 19.8 m. The test section where the ship model was installed has dimensions of 12.5 m (L) × 2.8 m (W) × 1.8 m (H). Figure 1a,b show an experimental setup for the current test ship model with full appendages in the LCT test section. The ship hull form is a surface combatant, and its model size has a length of 7.07 m, a breadth of 0.933 m, and a draft of 0.269 m as shown in Table 1. The propeller is a controllable pitch propeller (CPP) with five blades and rotates outboard. The propeller model diameter ($D_P$) is 0.28 m. As shown in Table 1, the flat rudder has the section profiles of symmetrical NACA sections at the root and the tip. As will be explained in detail later, three twisted rudders were made by rotating each cross-section of the flat rudder around the mid-chord point according to the predicted inflow angle distributions. The chord length (*c*) of the rudder model is 0.1967 m at the root and 0.0887 m at the tip. The span length (*s*) is 0.2147 m. The sign of the rudder angle, $\delta_R$, was defined as positive when rotating in the direction of the starboard side. Figure 1c shows the holes on the inboard and outboard surfaces of the flat rudder to measure the pressure distributions along the two-chord lines at the span locations, $z/s$ = 0.45 and 0.6. This pressure measurement was carried out at 18-knot ship speed and non-cavitating flow conditions. Fourteen pressure taps were installed along the rudder surface at 60% span of the rudder. The relative pressure was measured with the pressure transducers made by Validyne DP15. The pressure was measured with an uncertainty of 0.18%. The pressure was non-dimensionalized as follows:

$$C_p = \frac{p - p_o}{\frac{1}{2}\rho_w U_o^2} \tag{1}$$

where, $p$ is the pressure, $p_o$ is the free stream reference pressure, $\rho_w$ is the water density and $U_o$ is the inflow velocity in the LCT test section. The hydrodynamic performances of the flat and twisted rudders were estimated by the surface cavitation, the drag and lift

forces, and the moment for each rudder. For this purpose, the model tests were performed for cavitating flows at a relatively high ship speed, $V_s$ = 30 knots. The corresponding test conditions are shown in Table 2. Here, $K_T$ is the thrust coefficient, $n$ is the propeller rotational speed, $\sigma_{n,0.5R_P}$ is the cavitation number at the radial position of $0.5R_P$ and $R_P$ is the propeller radius. The thrust coefficient $K_T$ and the cavitation number $\sigma_{n,0.5R_P}$ are given as:

$$K_T = \frac{T}{\rho_w n^2 D_P^4} \tag{2}$$

$$\sigma_{n,0.5R} = \frac{p_T - p_v}{0.5\rho_w n^2 D_P^2} \tag{3}$$

where, $T$ is the thrust of the propeller, $p_T$ is the static pressure in the LCT and $p_v$ is the vapor pressure. The rudder forces and moment were measured by using a dynamometer. A dynamometer was installed in the stern hull over the rudder to measure the drag force ($\mathcal{D}$) in the x-direction, the lift force ($\mathcal{L}$) in the y-direction, and the moment ($\mathcal{M}$) in the z-direction acting on the rudder. The dynamometer was manufactured by Wonbang Forcetech Co. Ltd., Daejeon, Korea, and can measure drag, lift, and moment up to 1000 N, 2000 N, and 45 Nm, respectively with uncertainties of 0.38%, 0.23%, and 1.70%. The drag, lift, and moment coefficients are defined as follows:

$$C_D = \frac{\mathcal{D}}{\frac{1}{2}\rho_w U_o^2 S} \tag{4}$$

$$C_L = \frac{\mathcal{L}}{\frac{1}{2}\rho_w U_o^2 S} \tag{5}$$

$$C_M = \frac{\mathcal{M}}{\frac{1}{2}\rho_w U_o^2 S \ell_c} \tag{6}$$

where, $S$ is the lateral projected area of the rudder and $\ell_c$ is the moment arm from the rudder centroid to the rotation axis in the chord direction. Surface cavitation on each rudder was recorded through a charge-coupled device (CCD) video camera with the accompanying lightening system.

**Table 1.** Main dimensions of the ship, propeller and rudder models.

|           |                            |          |
| --------- | -------------------------- | -------- |
| Ship      | Length between perpendiculars | 7.067 m  |
|           | Breath                     | 0.933 m  |
|           | Draft                      | 0.269 m  |
| Propeller | Propeller Diameter         | 0.28 m   |
|           | Number of blades           | 5        |
|           | Rotation Direction         | Outward  |
| Rudder    | Root chord length          | 0.1967 m |
|           | Tip chord length           | 0.0887 m |
|           | Span length                | 0.2147 m |

**Table 2.** LCT test conditions for cavitating flow.

| $V_s$      | $K_T$  | $\sigma_{n,0.5R}$ | $U_o$     | $n$       |
| ---------- | ------ | ----------------- | --------- | --------- |
| 30.0 kts   | 0.1879 | 1.2485            | 9.0 m/s   | 31.61 rps |

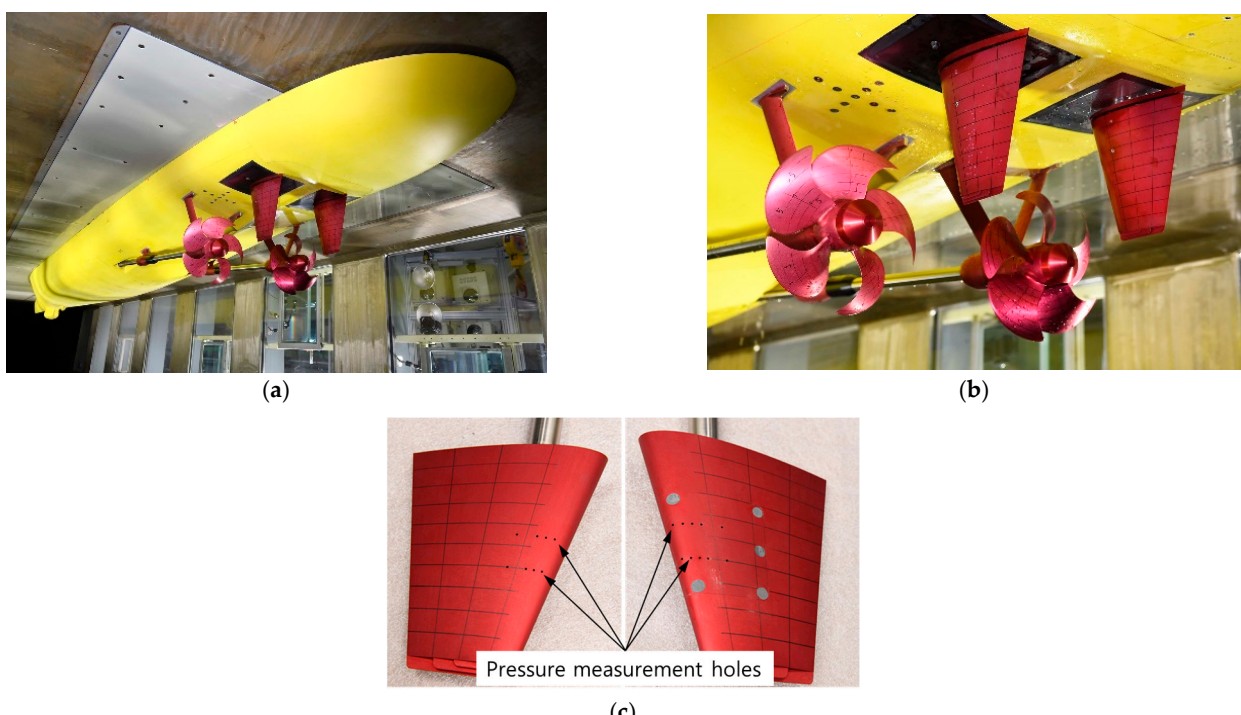

**Figure 1.** Experimental setup in the LCT: (**a**) full appended ship model mounted in the test section; (**b**) close-up view of the stern region; (**c**) pressure measurement holes on the inboard and outboard surfaces of the port flat rudder.

## 3. Numerical Approaches

### 3.1. Reynolds-Averaged Navier-Stokes Equations

The RANS equations with turbulence models were solved to simulate wetted and cavitating flows around the test ship model, since the RANS models have been shown to be still successful in the estimation of the hydrodynamic performances of ships and offshore structures. A single set of the mass conservation and RANS momentum conservation equations in Cartesian coordinates for two-phase (vapor and liquid) incompressible flows is given by:

$$\frac{\partial U_i}{\partial x_j} = 0 \tag{7}$$

$$\frac{\partial (\rho U_i)}{\partial t} + U_j \frac{\partial (\rho U_i)}{\partial x_j} = -\frac{\partial P}{\partial x_j} + \frac{\partial}{\partial x_j}\left(\mu \frac{\partial U_i}{\partial x_j}\right) + \frac{\partial \left(-\rho \overline{u_i' u_j'}\right)}{\partial x_j} \tag{8}$$

where, $\rho$ is the fluid density, $U_i$ are the time-averaged velocity components corresponding to the Cartesian coordinates $x_i$, $t$ is time, $u_i'$ are the velocity fluctuations, $P$ is the time-averaged pressure, $\mu$ is the fluid dynamic viscosity and $-\rho \overline{u_i' u_j'}$ are the Reynolds stresses. The Reynolds stress term can be defined in terms of known quantities. In turbulence modeling, the problem is to develop a suitable closure model to predict the Reynolds stresses. As one of the various types of turbulence closure models, the Boussinesq hypothesis provides a simple relationship between the Reynolds stresses and velocity gradients through the eddy viscosity $\mu_t$ as follows:

$$-\rho \overline{u_i' u_j'} = \mu_t \left(\frac{\partial U_i}{\partial x_j} + \frac{\partial U_j}{\partial x_i}\right) \tag{9}$$

Among the two-equation turbulence models, the realizable $k - \varepsilon$ model based on the suggestions by Shih et al. [40] was used. This model uses the transport equations for the turbulent kinetic energy $k$ and turbulence dissipation rate as the standard $k - \varepsilon$ model. Better predictions of the distribution of the dissipation rate and boundary layer

characteristics in a large pressure gradient, separated, and recirculating flows can be expected compared to the standard $k - \varepsilon$ model.

### 3.2. Cavitation Model

A mixture approach was used to solve the cavitating flows around a hydrofoil, which assumes that the flow is a homogeneous vapor-liquid mixture, and the vapor is homogeneously distributed in a finite volume of liquid. Based on this assumption the mixture fluid can be treated as a single pseudo-fluid with variable fluid properties corresponding to the composition of two fluids. The density and viscosity of mixture fluid are averaged on a volume fraction basis as follows:

$$\rho = \alpha_v \rho_v + (1 - \alpha_v)\rho_l, \ \mu = \alpha_v \mu_v + (1 - \alpha_v)\mu_l \tag{10}$$

where, $\alpha$ is the fluid volume fraction and the subscripts $v$ and $l$ indicate the vapor and liquid phase, respectively. The continuity equation for the vapor volume fraction with the mass transfer rate $\dot{m}$ from the vapor to the liquid can be written as,

$$\frac{\partial}{\partial t}(\alpha_v \rho_v) + \frac{\partial}{\partial x_j}(\alpha_v \rho_v u_j) = -\dot{m} \tag{11}$$

The modeling of a suitable mass transfer rate is the key for a cavitation model. The model proposed by Schnerr and Sauer [41] which had been adopted in Star-CCM+ was used in the simulations of cavitating flows, which can be expressed as

$$\dot{m} = 3\frac{\rho_v \rho_l}{\rho_m} \frac{\alpha(1-\alpha)}{R} \sqrt{\frac{2}{3}\frac{|p - p_v|}{\rho_l}} sgn(p_v - p) \tag{12}$$

In this model, the vapor fraction is described by $N$ spherical bubbles of radius $R$ and the nuclei concentration per unit volume of liquid, $n_o$, as follows:

$$\alpha = \frac{n_0 (4/3)\pi R^3}{1 + n_0 (4/3)\pi R^3} \tag{13}$$

The Schnerr-Sauer cavitation model uses a reduced Rayleigh–Plesset equation, which neglects the influence of surface tension, bubble growth acceleration, viscous effects, and the bubble-bubble interaction. Otherwise, this model includes scaling of the bubble growth and collapse rates for both single-component and multi-component materials. The cavitation bubble growth rate can be calculated through the simplified Rayleigh relation as follows:

$$\frac{dR}{dt} = \sqrt{\frac{2}{3}\frac{|p - p_v|}{\rho_l}} sgn(p_v - p) \tag{14}$$

### 3.3. Numerical Solution Procedure

Figure 2 shows the computational domain of the full appended ship model which has the same test section of the LCT. The velocity inlet boundary was located at the starting point of the LCT test section, and the pressure outlet boundary was placed 1.5 model lengths downstream of the hull stern. The wall boundary condition was applied on the hull and other boundaries. Figure 2b shows the local computational domain rotating with the propeller. The governing equations for two-phase incompressible flows were solved in the commercial program Star-CCM+ by using the finite volume method and a volume of fluid approach [42]. The segregated flow approach was used to solve the equations in which the Semi-Implicit Method for Pressure-Linked Equations (SIMPLE) algorithm was adopted to resolve the pressure-velocity coupling. A second-order accurate upwind scheme was used for the spatial differencing of the convective terms. A second-order central differencing was used for the viscous terms. The solution procedure was separated into two steps in the

simulation of non-cavitating flows. In the first step, the steady computation of the propeller rotation was carried out using the moving reference frame (MRF) method. The second step involved the unsteady computation of the propeller rotation using a rigid body motion (RBM), the so-called sliding mesh method until a periodic convergence was achieved. Here, the computational time step corresponded to the time taken to rotate 2° at a given propeller rotation speed shown in Table 2. The cavitating flows were solved by employing a second-order time implicit scheme and 10 inner iterations to reach convergence at a given physical time, where the converged unsteady-state solutions for non-cavitating flows were utilized as the initial conditions. The time step for the cavitating flow simulations corresponded to 1° rotation time at a given propeller rotation rate.

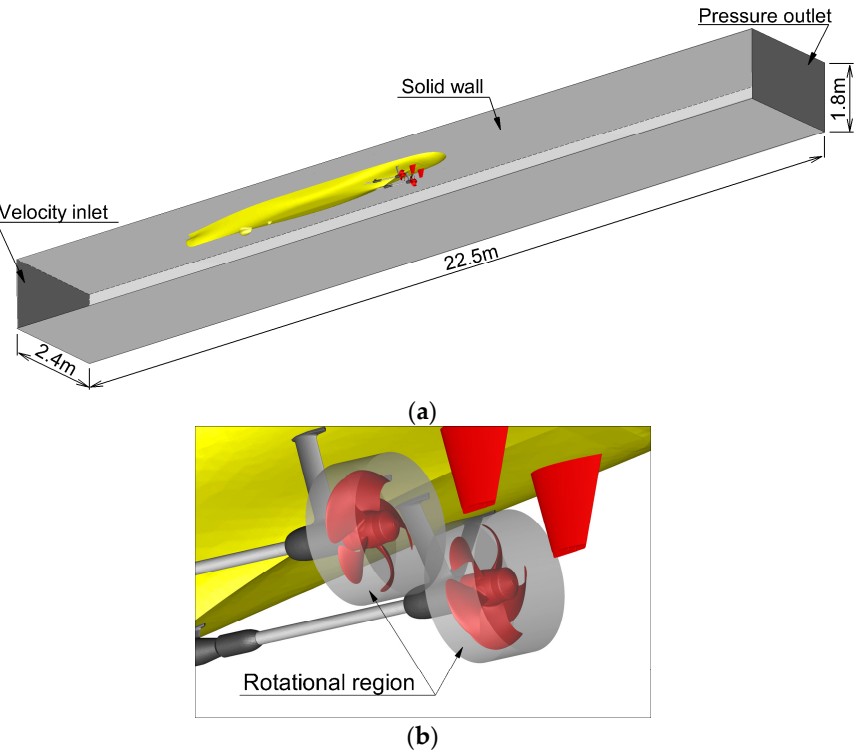

**(a)**

**(b)**

**Figure 2.** Computational domain: (**a**) computational domain of the ship model and boundary conditions; (**b**) starboard and port rotational regions containing the propellers.

Figure 3 shows the surface and field grids around the full appended hull. An unstructured grid based on hexahedral and polyhedral meshes topology was used in the present simulations. To see the grid convergence on rudder forces, three grids called coarse, medium, and fine with 5.2 M, 7.5 M, and 11 M cells, respectively, were generated by a refinement ratio of $\sqrt{2}$. A prism layer was used to generate orthogonal prismatic cells next to the wall surfaces, in which the height of the first cells from the wall was determined so that the dimensionless wall distance $y^+$ was set to be 70 or less. Here, all grids had 11 prism grid layers to have a proper resolution of the boundary layer around the hull, propeller, and rudder. Especially, an overset grid was adopted to consider the rotations of each rudder in the starboard and port side at a given rudder angle. The present computations were carried out on a Linux-based PC cluster system whose each node has 20 Intel®_Xeon® 2.40 GHz processors. All runs used a total of 280 processors.

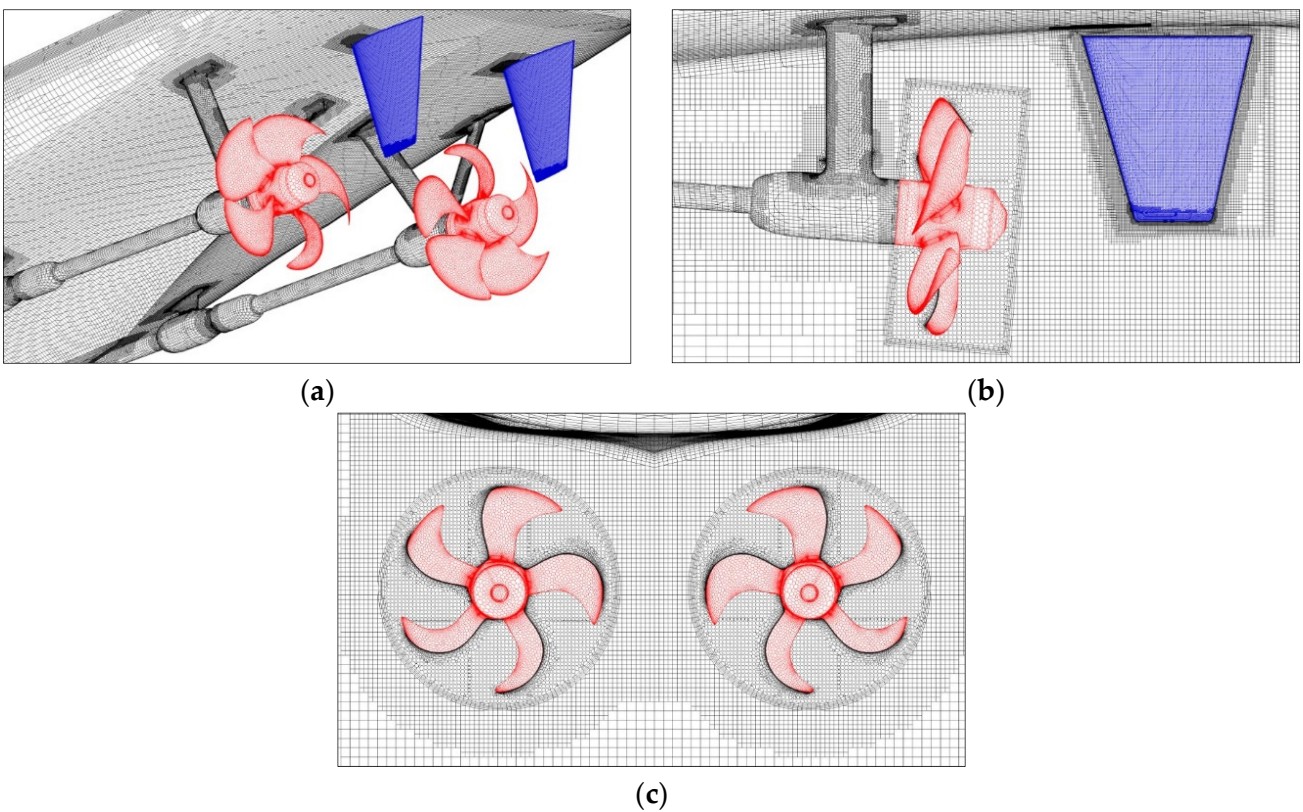

**Figure 3.** Computational surface and field grid distributions around the hull, propeller, and rudder: (**a**) grid distribution on the full appended hull surface; (**b**) field grids on the longitudinal plane passing through the center of the port rudder; (**c**) field grids on the transverse plane passing through the center of the propeller.

## 4. Results and Discussions

### 4.1. Validation of the Flow around a Flat Rudder

4.1.1. Pressure Distributions

The measured and calculated pressure distributions for wetted flow along the two-chord lines of the port side (PRT) flat rudder are shown in Figure 4 at the rudder angle of $\delta_R = 0°$ and a ship speed of 18 knots. Due to the angle of attack of the flow into the rudder induced by the propeller, it was observed that the inboard of the rudder became the pressure side, and the outboard corresponded to the suction side. It is seen that the pressures on the inboard and outboard surfaces differ more significantly at the span location of $z/s = 0.6$ as seen in Figure 4b, which means the inflow angle varies at each spanwise location at a given rudder angle condition. For a surface combatant having rudder geometry and propeller-rudder configuration similar to the test model proposed by Shen et al. [32], the inflow angle distribution showed a maximum value at the span location of about $z/s = 0.66$. The minimum pressure of the outboard surface is rather large even at $\delta_R = 0°$ and increased significantly with an increase in the rudder angle. This result indicates that cavitation inception and cavitation erosion are expected to occur at higher rudder angles. The influences of the inflow angles can be effectively compensated by applying different twist angles in the spanwise direction of a rudder. The current numerical calculations were obtained by solving the hull-propeller-rudder interaction of the target ship model under the same conditions of the LCT model tests, showing a good agreement with the measured pressure distributions. This indicates that the numerical approach used in the present study is able to adequately estimate rudder performances. Although the overall difference between the grids used in this result is not large as seen in the figure, the fine grid shows a more reasonable agreement.

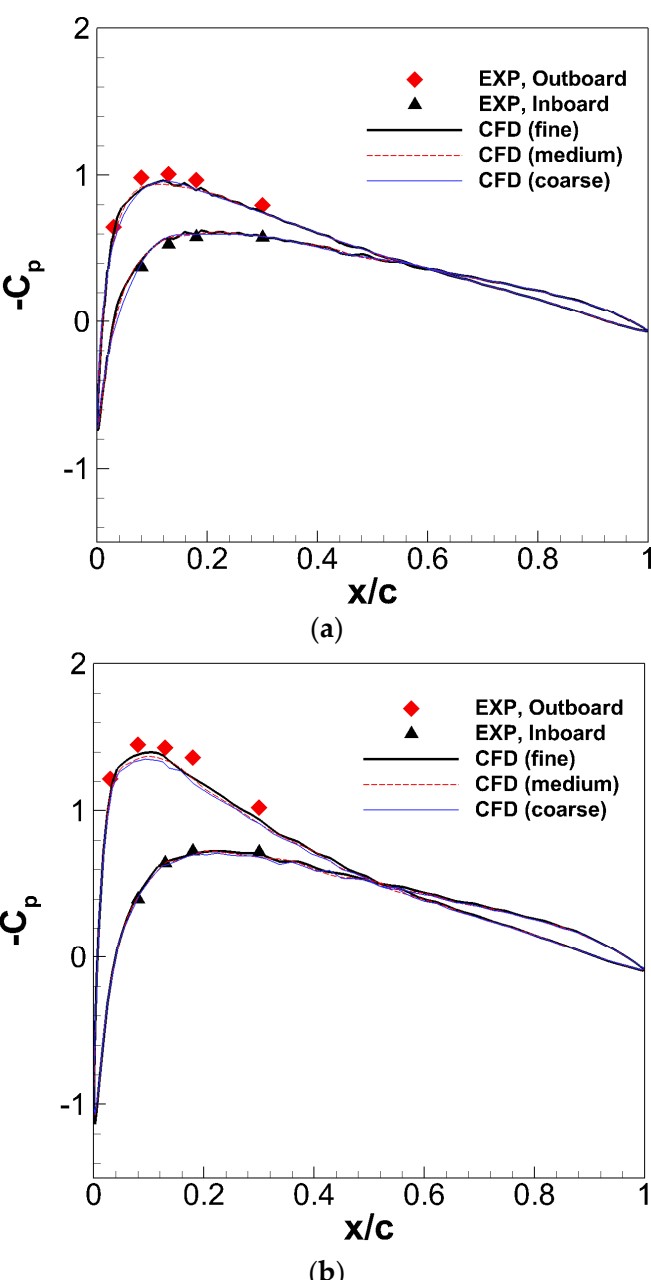

**Figure 4.** Pressure distributions along the chord lines on the flat rudder: (**a**) *z/s* = 0.45; (**b**) *z/s* = 0.6.

4.1.2. Surface Cavitation

Figure 5 shows the predicted surface cavitation on the flat rudder with that observed in the LCT model tests at the rudder angles of $\delta_R$ = 0°, −4°, −8° and −12° and a ship speed of 30 knots. Under all conditions, surface cavitation on the outer surface of the port rudder and the inner surface of the starboard side (STB) rudder is seen. The main difference between the numerical and experimental results was that in the experiment, complicated cloud-cavitation was formed downstream just after the sheet cavitation inception at the leading edge of the rudder. Since the present numerical study used the RANS equations as the governing equations of cavitating flows and the VOF method which is a grid-based Eulerian method for cavitation modeling, it was difficult to precisely capture the behavior of cloud cavitation of particle-like characteristics which vary over a short time. However, it can be noted that the current numerical results agree qualitatively with the experimental results regarding the range and amount of cavitation occurrence according to the change in rudder angle. As expected from the previous results of the pressure distributions along

the chord lines, the surface cavitation inception is seen around the region of $z/s = 0.7$ even at $\delta_R = 0°$. From this cavitation inception, the rudder cavitation significantly expanded in the chord and span directions as the rudder angle increased. In addition, the tip cavitation inception was detected at the leading edge at $\delta_R = 0°$. The current study also tried to minimize this kind of cavitation by modifying the tip geometry. This result will be shortly discussed later.

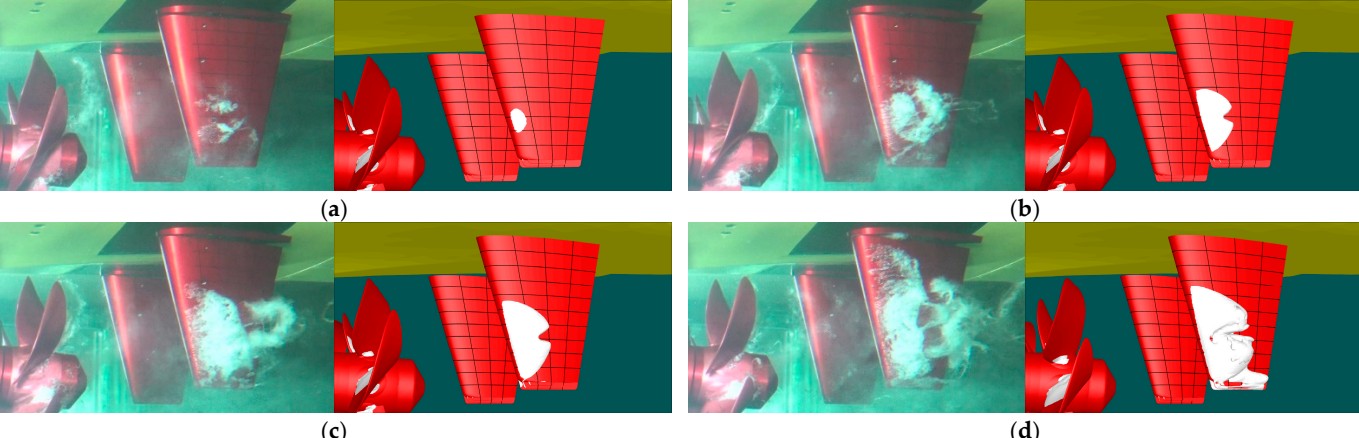

**Figure 5.** Experimental observation and numerical prediction of surface cavitation on the twin flat rudders: (**a**) $\delta_R = 0°$; (**b**) $\delta_R = -4°$; (**c**) $\delta_R = -8°$; (**d**) $\delta_R = -12°$.

### 4.1.3. Forces and Moment

The measured and predicted lift and drag forces and moment for the flat rudder at 30 knots and cavitating flow condition are shown in Figure 6. The moment was measured with respect to the rudder stock axis that is the center of rotation of the rudder. In Figure 6a, the lift coefficients are seen to linearly change when both starboard and port rudders have no or slight cavitation occurrence. At rudder angles larger than $\delta_R = 8°$ in absolute value for each side of the twin rudder with significant cavitation, it is seen that the lift forces vary nonlinearly and decrease somewhat because of massive cavitation occurrence. The zero-lift angles are placed at around $-4.69°$ and $4.91°$, respectively, which was due to the inflow angles induced by the propeller. The zero-lift angle can be expected to become around $0°$ for a twisted rudder applied with appropriate inflow angles. The trend of this measured lift force is closely followed by the current numerical results. The difference between the experimental and numerical results was caused by the limitation of the current numerical approach where cloud type cavitation occurs as discussed earlier. For each rudder with surface cavitation, the predicted lift forces are smaller than the experimental values because the predicted cavitation was nearly attached to the rudder surface unlike the cloud type cavitation which detached off the rudder surface, as observed in the experiment, and increased the suction side pressures. In Figure 6b,c, similar trends are also seen in the comparisons of the drag force and moment. Where the cavitation effect is not significant, the predicted forces and moment show a very good agreement with the experimental data. When the grid dependence in the results for the rudder angles of $\delta_R = 0°$, $8°$, and $16°$ was examined, the difference between the grids was not significant. From these results, it can be noted that the current numerical approach shows a reasonable accuracy enough to qualitatively estimate the hydrodynamic forces of rudders.

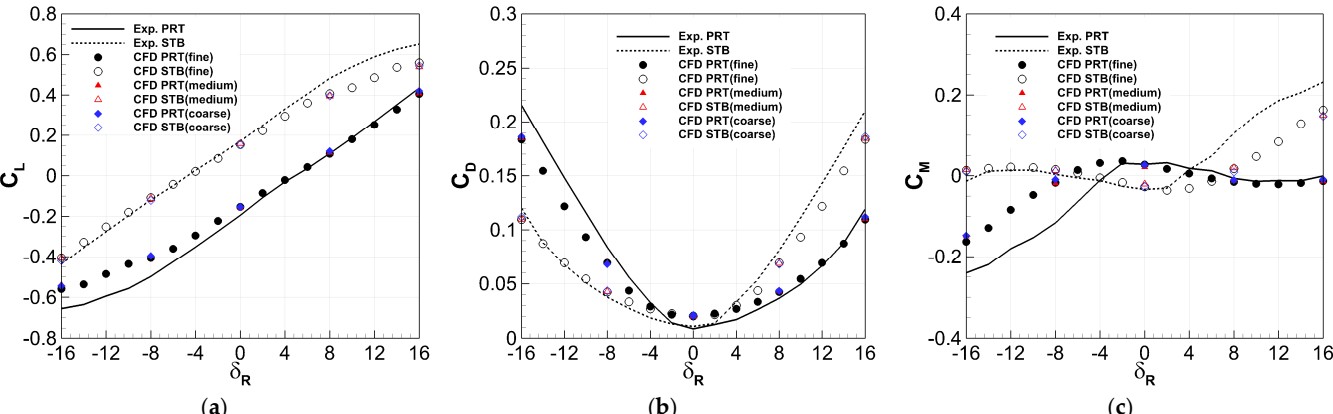

**Figure 6.** Measured and numerically predicted rudder forces and moment for the twin flat rudders: (**a**) lift force; (**b**) drag force; (**c**) moment.

### 4.2. Prediction and Correction of Twist Angles

The predicted instantaneous axial and transverse velocity distributions on the longitudinal plane passing the nose-tail line of the port flat rudder at $\delta_R = 0°$ and 30-knot ship speed are shown in Figure 7. Excluding the region below the rudder, it is seen that the accelerated axial flow occupies about 50–100% region of the span from the root. The flow velocities decreased near the leading edge due to the influence of the wall boundary condition. In the same region, due to the propeller rotation, the distribution of the relatively large transverse velocity is seen to be in the negative y-direction, which means the port rudder will have negative inflow angles along the span direction at $\delta_R = 0°$. These results roughly show where the onset flow angles induced by the propeller will be significant.

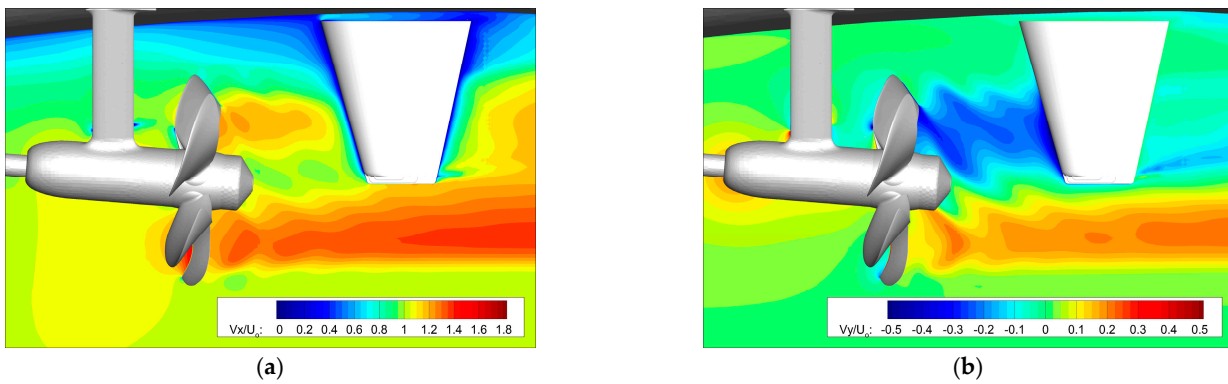

**Figure 7.** Instantaneous axial and transverse velocity distributions on the longitudinal plane passing the nose-tail line of the port side flat rudder at $\delta_R = 0°$ and 26 knot ship speed: (**a**) axial velocity distribution; (**b**) transverse velocity distribution.

Figure 8a,b show the time-averaged distributions of axial and transverse velocity components, respectively, in front of the rudder at $\delta_R = 0°$ on the same plane shown in Figure 7. The profiles of the axial and transverse velocities along an inclined line parallel to the leading edge of the rudder between the propeller and rudder are shown in Figure 8c. The rudder span region where the inflow angle is significant is shown to be clearer. The location of the maximum inflow angle was detected in the region of $z/s = 0.6\sim0.8$, where the magnitudes of the axial and transverse velocities are larger than their surroundings within the rudder span. The profiles of the velocity components compared with the relative positions of the propeller and rudder seen in Figure 8c show the magnitudes of each inflow velocity component entering the rudder placed behind the wake of the propeller. The humps in the velocity profiles over the rudder region were due to the effects of the accelerated and rotating flow caused in the upper half of the propeller.

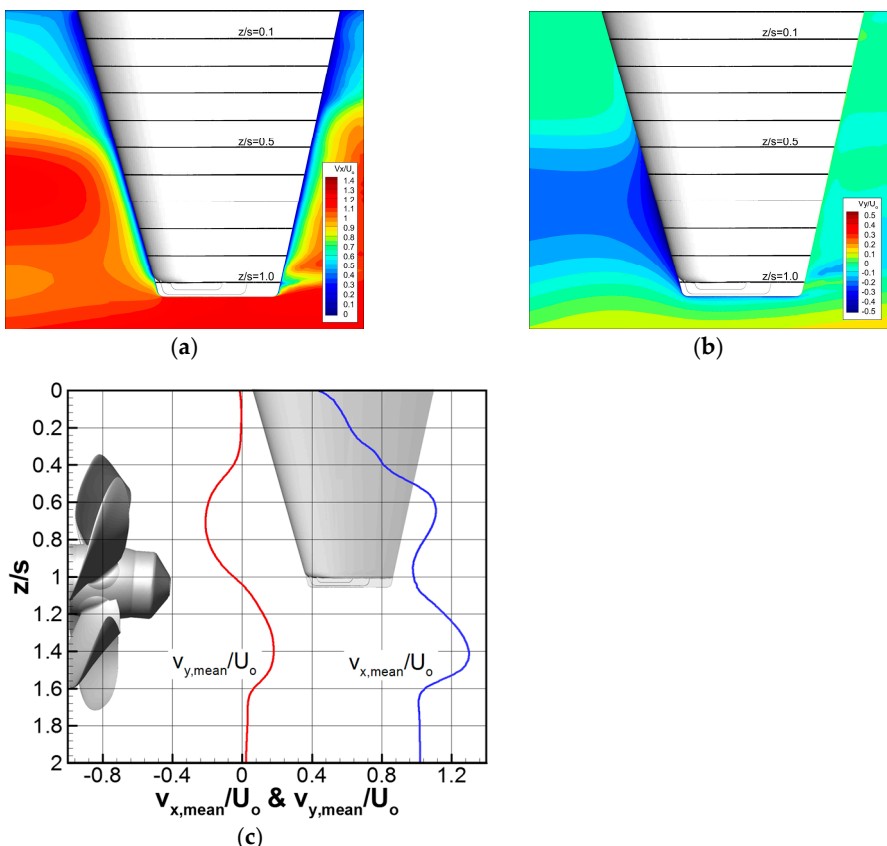

**Figure 8.** Time-averaged distributions of axial and transverse velocity components and the profiles of those two velocities in front of the leading edge of the flat rudder: (**a**) axial velocity distribution; (**b**) transverse velocity distribution; (**c**) axial and transverse velocity profiles.

Figure 9 shows a transverse plane between the propeller and the rudder aligned with the inclined leading edge of the flat rudder keeping a distance of $0.26D_P$ in which velocity distributions of the axial and transverse components were measured to obtain the inflow angles.

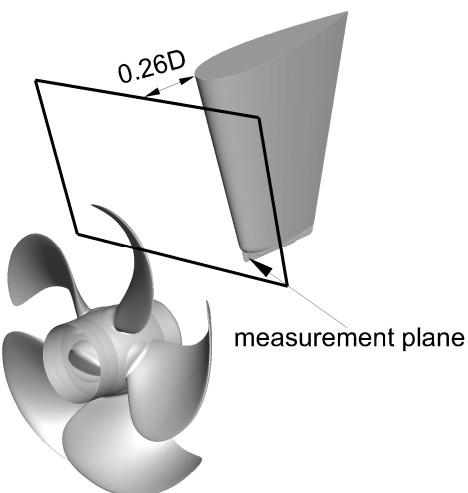

**Figure 9.** Measurement plane location in which inflow angles were predicted using the axial and transverse velocities.

Figure 10 shows the numerically predicted flow angle distribution on the measurement plane shown in Figure 9. Due to the propeller rotation, the onset flow angle has opposite

signs at the upper and lower propeller planes as seen in Figure 10a. The maximum inflow angles with opposite signs were observed around the radius of $0.56R_P$ in the 7 o'clock and 1 o'clock positions, respectively. The predicted inflow angles along the rudder span direction for the rudder angle range of $\delta_R = -16°-16°$ are shown in Figure 10b. The distribution of the inflow angles along the rudder span direction is in the range of $0°-11°$ for the rudder angle of $\delta_R = 0°$. The range of the maximum inflow angle experienced by the rudder at the rudder angle range of $\delta_R = -16°-16°$ corresponds to about $5°-15°$.

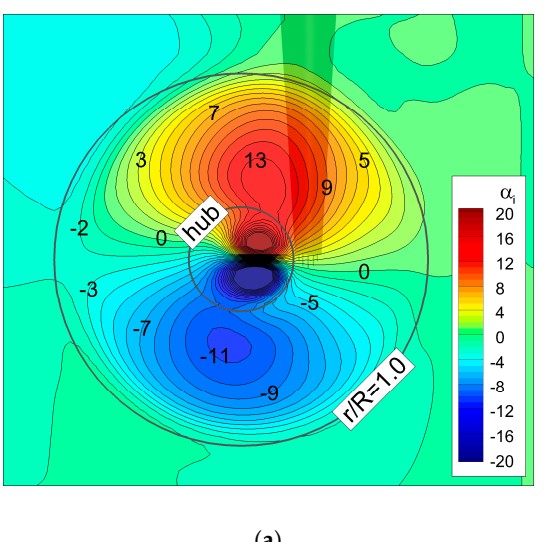

(**a**)

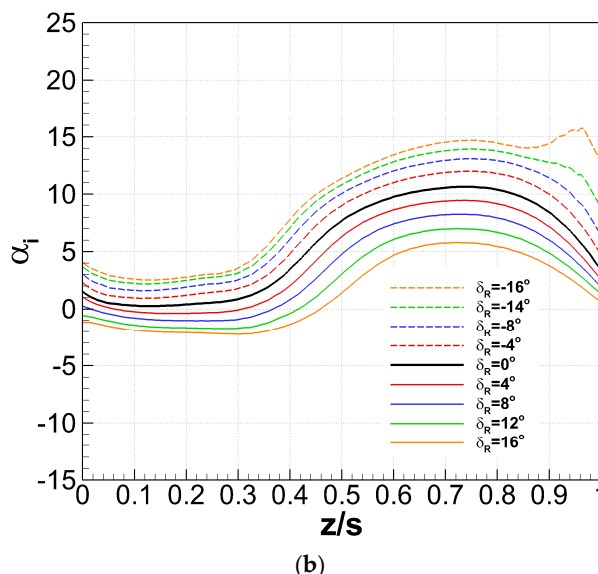

(**b**)

**Figure 10.** Numerically predicted inflow angle distribution on a transverse plane in front of the port flat rudder and comparison of inflow angles predicted and measured according to rudder angle: (**a**) inflow angle distribution on the measurement plane shown in Figure 9; (**b**) inflow angles in the span direction according to rudder angle.

The initial twisted rudder model was designed based on the predicted inflow angle distribution along the rudder span at $\delta_R = 0°$. However, it was found that the twist angles can be overpredicted when extracted from the leading edge lines projected onto the measurement plane shown in Figure 9. When the streamline distribution on the cross-sections in the span direction of the port rudder, as shown in Figure 11, was examined, it was found that the inflow angles computed at the initial prediction points should be corrected. At the selected span locations of $z/s = 0.3$, 0.5, 0.7, and 0.9, the streamlines passing through the initial measurement points are seen to be slightly biased toward the outboard direction of the port rudder which leads to a slight overestimation of the predicted initial inflow angles. The correction of the flow angle distribution was done by searching the streamline that had the stagnation point on the leading edge of the rudder. In this work, the maximum inflow angle was estimated to be about 3° smaller.

Figure 12 compares the twist angles before and after correction along the rudder span direction. The symbols show the inflow angles calculated at each span position, and the lines are the results of the interpolation of those angles using polynomial curves. The intermediate twist angle distribution was extracted to validate the final corrected twist angles by applying them together to the numerical and experimental analyses. The corrected twist angle distribution had a range from 0° to about 8.1°, while the intermediate twist angles were in a range from 0° to about 9.2°.

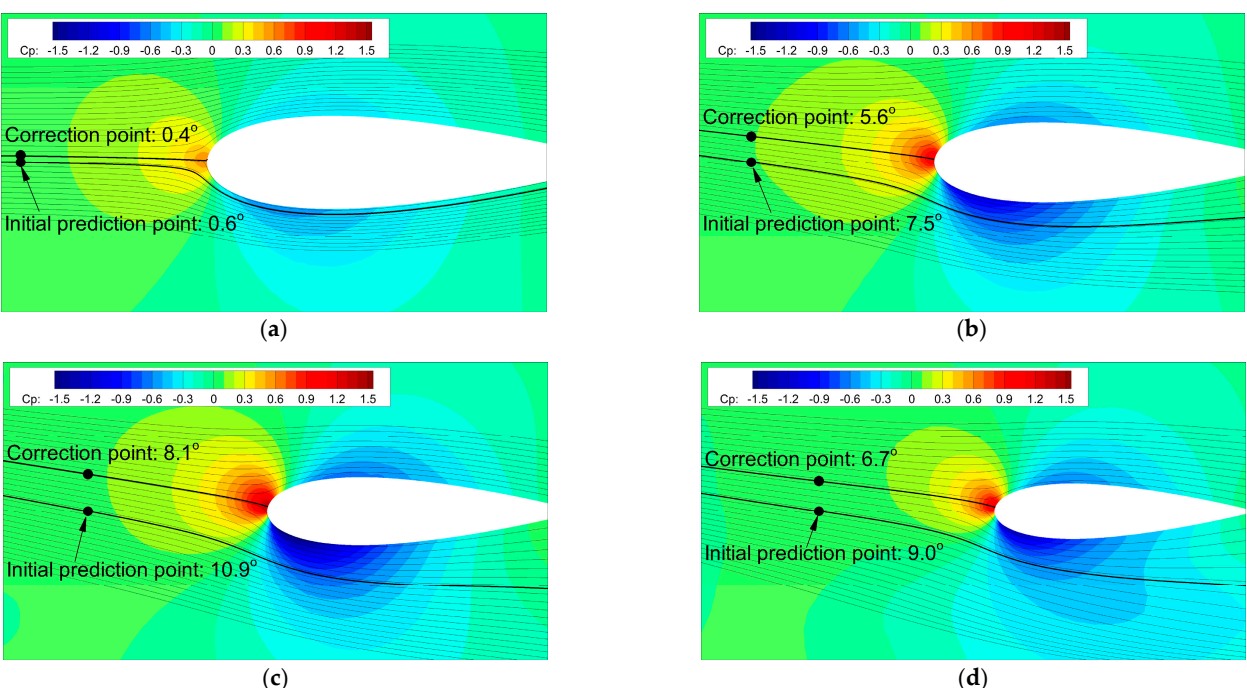

**Figure 11.** Prediction and correction points at which inflow angles were calculated by using the axial and transverse velocities: (**a**) $z/s$ = 0.3; (**b**) $z/s$ = 0.5; (**c**) $z/s$ = 0.7; (**d**) $z/s$ = 0.9.

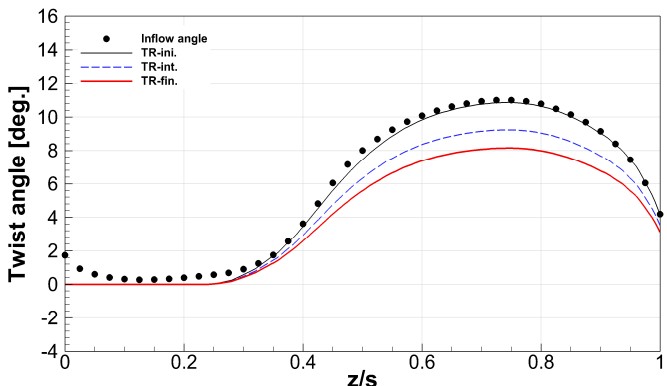

**Figure 12.** Distributions of twist angles along the span direction.

### 4.3. Hydrodynamic Performances of Twisted Rudders

Three twisted rudders shown in Figure 13 were made respectively by rotating each cross-section of the flat rudder around the mid-chord point according to the initial, intermediate, and final corrected twist angles described in Figure 12. Afterward, the three twisted rudders were named TR-ini, TR-int, and TR-fin. To examine the cavitation performance of the tip area, the TR-ini and TR-int twist rudders were attached with an end plate and a rounded tip was tested for the TR-fin model. For the investigation of the hydrodynamic performance of the twisted rudders, the numerical simulations of cavitating flows were carried out at 30-knot ship speed and the same propeller rotation condition used for the flat rudder as shown in Table 2.

The effectiveness of the twisted rudder TR-fin on the compensation of the onset flow angles was confirmed by recalculating the inflow angles from the distribution of the streamlines over each cross-section in the rudder span direction at $\delta_R = 0°$ as shown in Figure 14. While the inflow angles are seen to be slightly overcompensated by the TR-ini rudder, nearly zero angles of attack and symmetrical pressure distributions are

shown for the TR-fin rudder. The intermediate version, TR-int rudder presented the next best improvement.

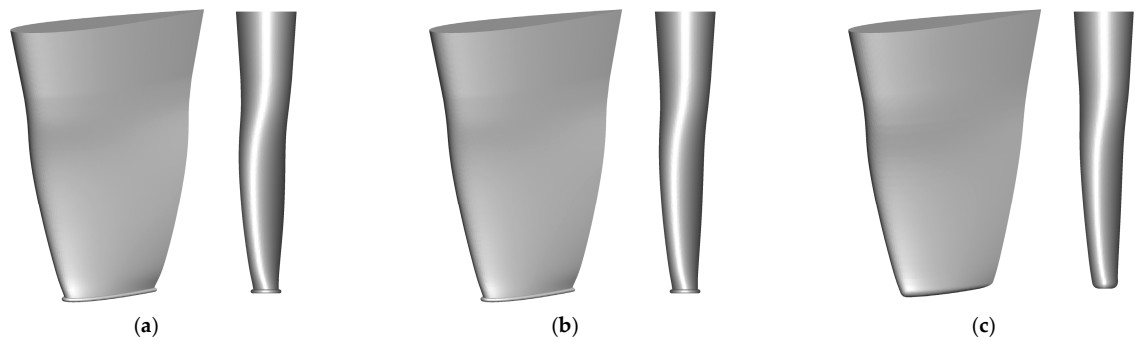

**Figure 13.** Designed twisted rudders: (**a**) TR-ini; (**b**) TR-int; (**c**) TR-fin.

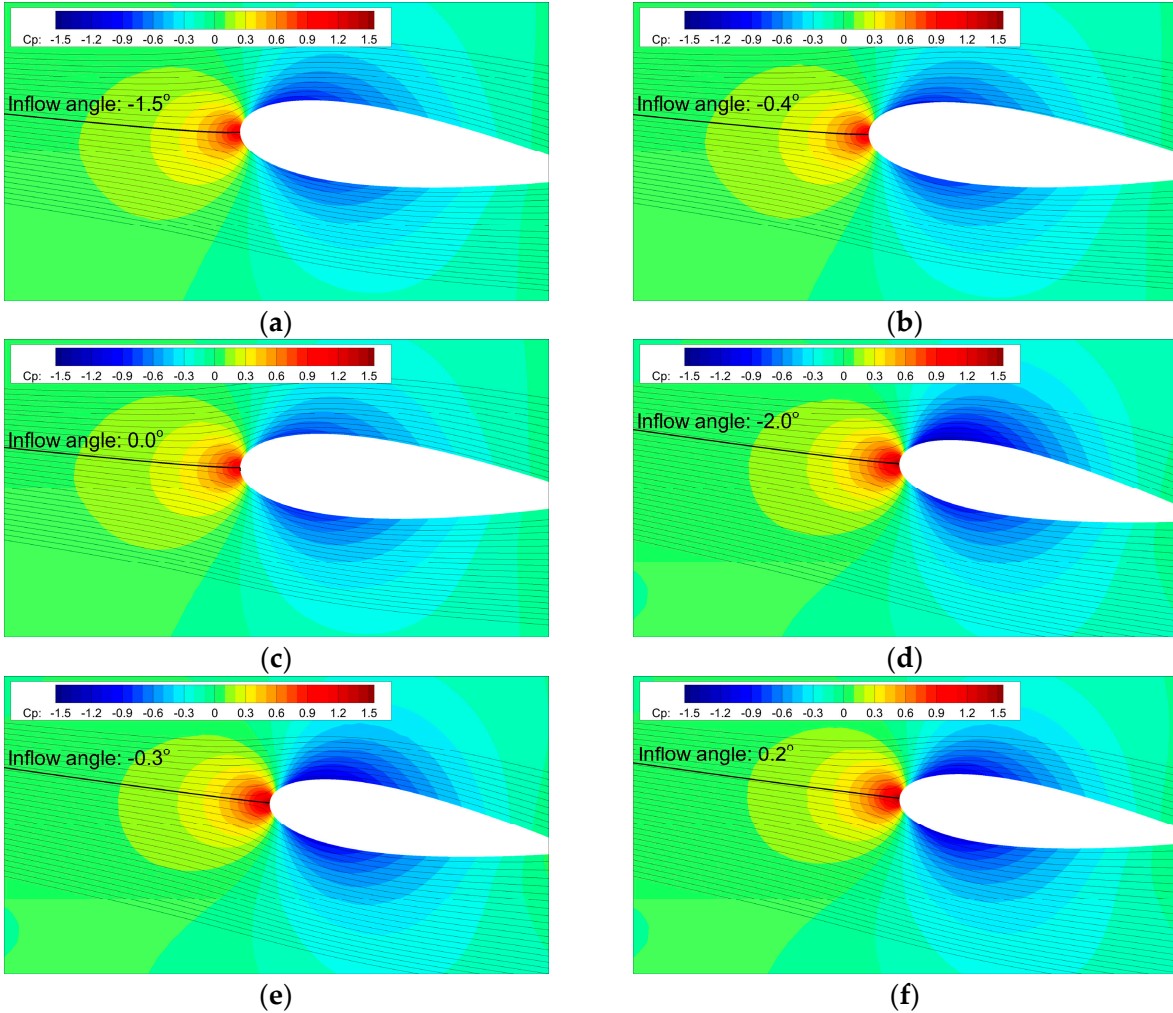

**Figure 14.** Streamline and pressure distributions on the cross sections in the span direction of three twisted rudders: (**a**) TR-ini at $z/s$ = 0.5; (**b**) TR-int at $z/s$ = 0.5; (**c**) TR-fin at $z/s$ = 0.5; (**d**) TR-ini at $z/s$ = 0.7; (**e**) TR-int at $z/s$ = 0.7; (**f**) TR-fin at $z/s$ = 0.7.

Figure 15 shows the pressure distributions along the rudder span direction from $z/s$ = 0.4–0.9 for the twisted rudders compared with those of the flat rudder. As seen in Figure 14, the pressure distributions for the TR-fin rudder are very close to each other on the inboard and outboard surfaces except for $z/s$ = 0.9, where the three-dimensional

flow effect near the tip region can be significant. The predicted pressure distributions for the flat rudder vary largely on each surface, especially at $z/s$ = 0.7~0.9, at which the cavitation inception appeared on the outboard surface. These results indicate that the surface cavitation inception phenomenon on the twisted rudder will be significantly improved compared with the flat rudder.

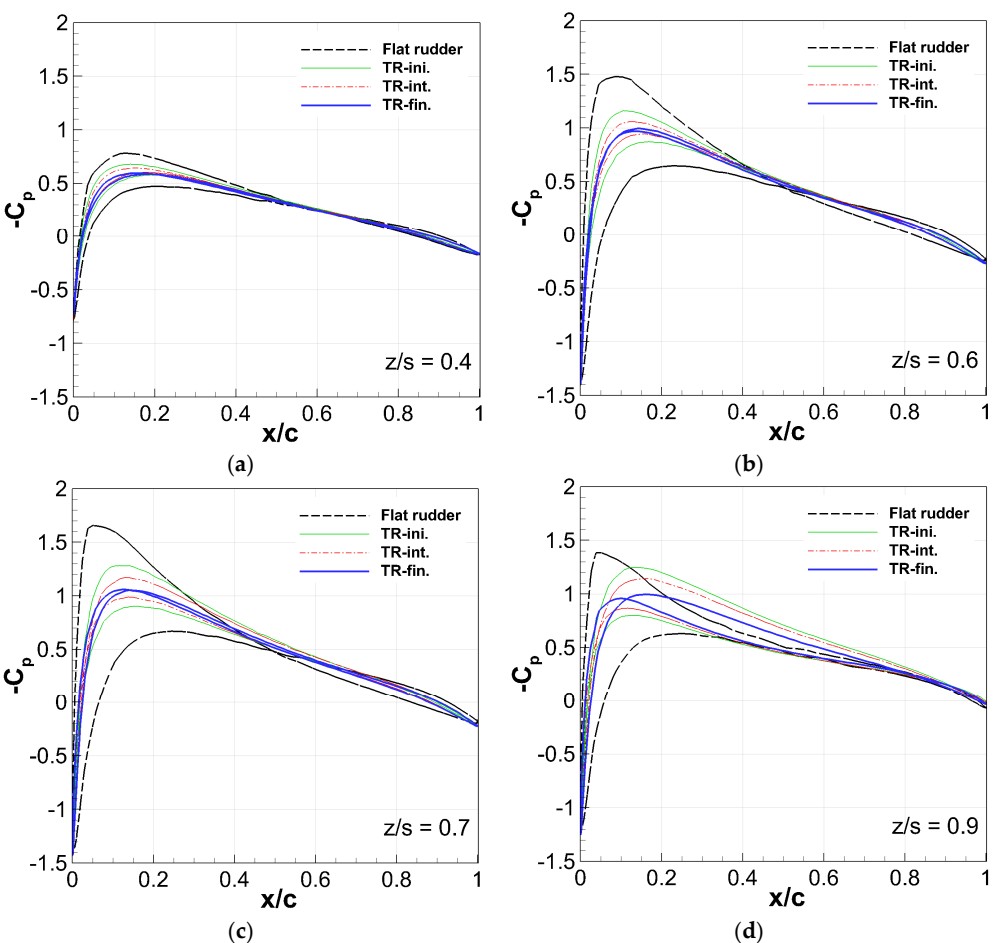

**Figure 15.** Pressure distributions along the chord lines on the twisted rudders: (**a**) $z/s$ = 0.4; (**b**) $z/s$ = 0.6; (**c**) $z/s$ = 0.7; (**d**) $z/s$ = 0.9.

Figures 16–18 show the predicted and observed surface cavitation on the flat and three twisted rudders at the rudder angles of $\delta_R$ = −6°, −8°, and −16°, respectively. As shown in Figure 5, the surface cavitation inception was detected even at $\delta_R$ = 0° for the flat rudder. For the twisted rudders, no surface cavitation was observed before $\delta_R$ = −6° in both the experiments and numerical simulations. While surface cavitation appeared on the outboard surface of the port flat rudder at $\delta_R$ = −6°, the same appeared on the inboard surfaces of each starboard twisted rudder. This was because each inflow angle distribution induced by the propellers on the starboard and port side was in opposite sign and both side rudders of a twin twisted rudder were made based on these twist angles. The starboard twisted rudder was unable to effectively compensate the inflow angles at negative rudder angles, and in the same way, the port twisted rudder was disadvantageous at positive rudder angles. As expected from the pressure distributions shown in Figure 15, it is seen that the cavitation occurrence is significantly decreased in the TR-fin rudder, followed by the TR-int and TR-ini rudders. Regarding tip cavitation inception, the rudder tip cavitation inception was also detected at the mid-chord of the tip on each inboard surface of both the starboard and port TR-ini and TR-int rudders equipped with a tip end plate at $\delta_R$ = 0°. On the other hand, such cavitation was observed on the TR-fin rudder above $\delta_R$ = −6°. This

indicates that the rounded tip applied to the TR-fin rudder is more useful than the end plate in reducing the tip cavitation.

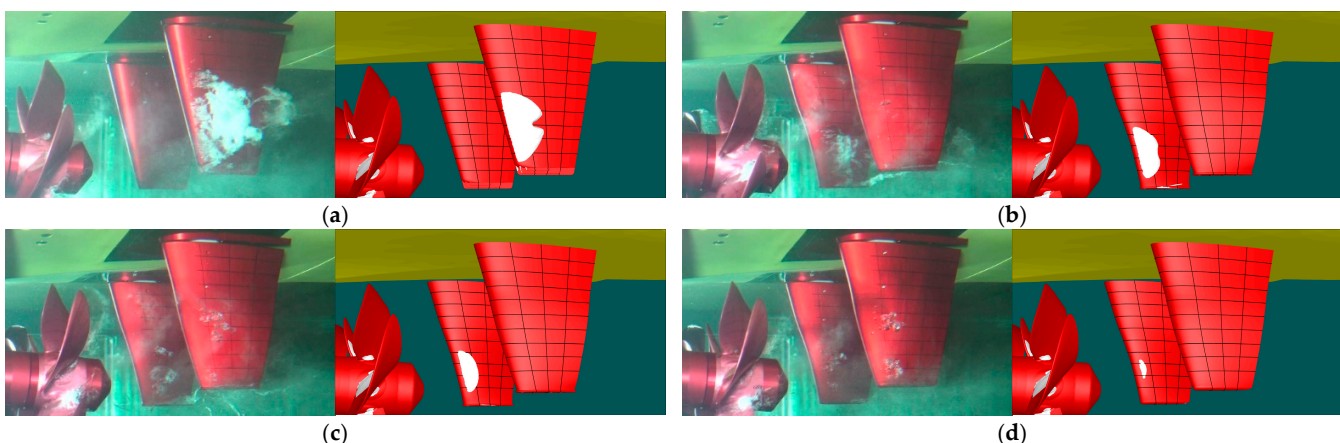

**Figure 16.** Experimental observation and numerical prediction of surface cavitation on the twin twisted rudders at $\delta_R = -6°$: (**a**) flat rudder; (**b**) TR-ini; (**c**) TR-int; (**d**) TR-fin.

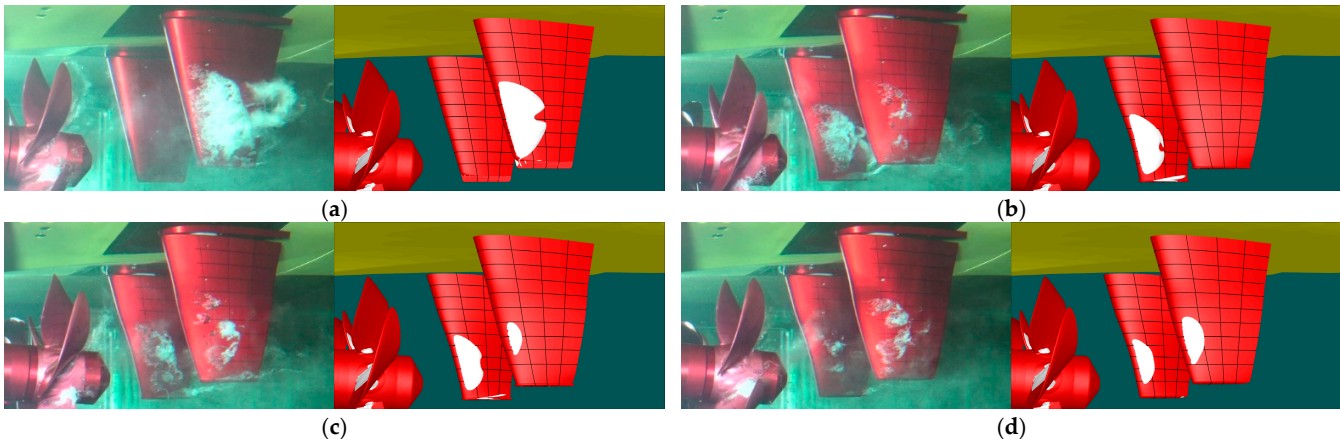

**Figure 17.** Experimental observation and numerical prediction of surface cavitation on the twin twisted rudders at $\delta_R = -8°$: (**a**) flat rudder; (**b**) TR-ini; (**c**) TR-int; (**d**) TR-fin.

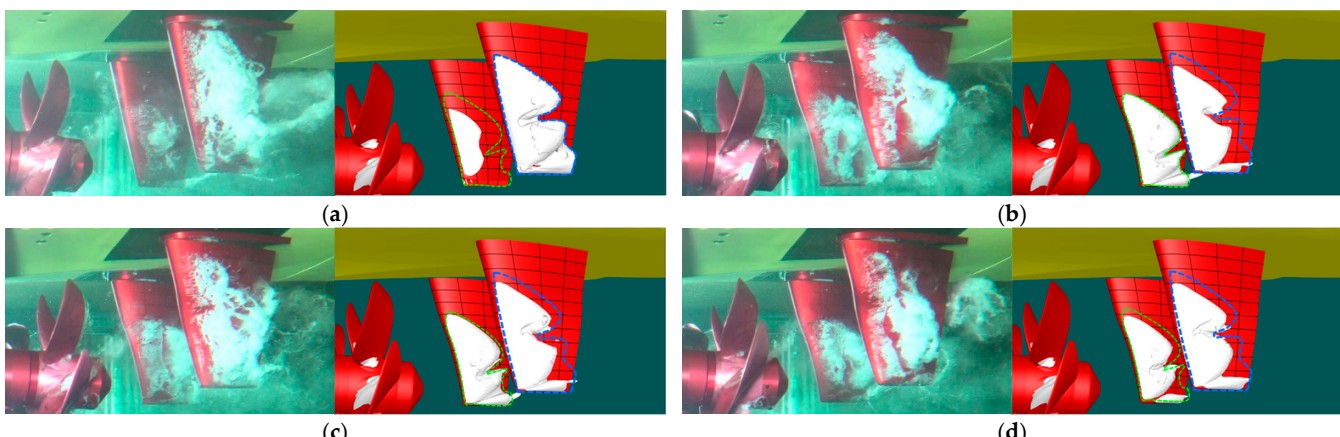

**Figure 18.** Experimental observation and numerical prediction of surface cavitation on the twin twisted rudders at $\delta_R = -16°$: (**a**) flat rudder; (**b**) TR-ini; (**c**) TR-int; (**d**) TR-fin.

It is seen in Figure 17 that the outboard surface cavitation begins to occur on each port twisted rudder at $\delta_R = -8°$. The inboard surface cavitation observed on the starboard twisted rudders at $\delta_R = -6°$ is slightly larger at $\delta_R = -8°$, but it is still the smallest on the TR-fin rudder.

At the rudder angle of $\delta_R = -16°$, severe surface cavitation is seen on all the rudder models as shown in Figure 18. For the flat rudder, it is seen that most of the outboard surface of the port rudder is covered with cavitation. Additionally, the inboard surface cavitation on the flat rudder started at $\delta_R = -14°$. To compare roughly the amount of the cavitation occurrence between the port rudders, the border line of the surface cavitation of the port flat rudder was overlaid on the outboard surfaces of the port twisted rudders as a dashed line. For the same purpose, the border line of the inboard surface cavitation of the starboard TR-ini rudder was superposed on the starboard TR-int and TR-fin rudders. The outboard surface cavitation on the port twisted rudders is seen to be largely decreased due to the compensation of the effects of the inflow angles. On the other hand, since a starboard twisted rudder at a negative rudder angle is disadvantageous in compensating the inflow angles, the inboard surface cavitation of the starboard twisted rudder can be increased than that on the flat rudder. For the twisted rudders, it is seen that the starboard surface cavitation on the TR-fin rudder is smaller than those on the other two twisted rudders. Therefore, these results show that the occurrence of surface cavitation in the TR-fin rudders on the starboard and port side is the smallest, showing the validity of the prediction and correction of the inflow angle distribution as described in the previous section.

The measured and predicted lift and drag forces and moment for the three twisted rudders at 30 knots and cavitating flow conditions are shown in Figure 19. As discussed for the results of the flat rudder in Section 4.1.3, the current numerical results for the twisted rudders show also a qualitatively good agreement with the experimental data. The minimum drag coefficients of each twisted rudder on the starboard and port side were found to be larger than that of the flat rudder at $\delta_R = 0°$ shown in Figure 6. In Figure 19a,d,g, the minimum positions are seen to be approximately $\delta_R = \pm 7°$, $\pm 6°$ and $\pm 5°$ for the TR-ini, TR-int, and TR-fin rudders, respectively. When considering the total drag of the starboard and port rudders, the total drag coefficients for the twin TR-ini and TR-int rudders were larger than that of the twin flat rudder in the current whole rudder angle range. In the numerical analysis, the total drag coefficient of the twin TR-fin rudders was smaller than that of the twin flat rudder above about $\delta_R = \pm 6°$, although this range was different from the experimental range of $\delta_R = \pm 4° \sim \pm 12°$.

With regard to the effect of the twist angles on rudder lift force, it is seen that each rudder lift force according to the rudder angle on the starboard and port side is almost similar for the TR-fin twisted rudder accounting for a relatively accurate inflow angle distribution as seen in Figure 19b,e,h. In the numerical simulation, the zero-lift angle of both sides of the TR-fin rudder is seen to be almost zero, but in the experiment, it is about 0.7° for the port rudder as seen in Figure 19h. The slope of the total lift force for each twisted rudder was similar to each other and slightly greater than that of the flat rudder. This means that a twisted rudder will be slightly better than a flat rudder in the maneuvering performance of ships.

Finally, it is seen that each rudder moment on the starboard and port side according to the rudder angle as seen in Figure 19c,f,i is getting closer to each other in the order the TR-fin, TR-int, and TR-ini rudders as the lift forces. The maximum moment coefficient was found to decrease in the same order. This shows that when an accurate twist angle distribution is applied, the rudder moment can be appropriately reduced without a significant reduction in the lift force.

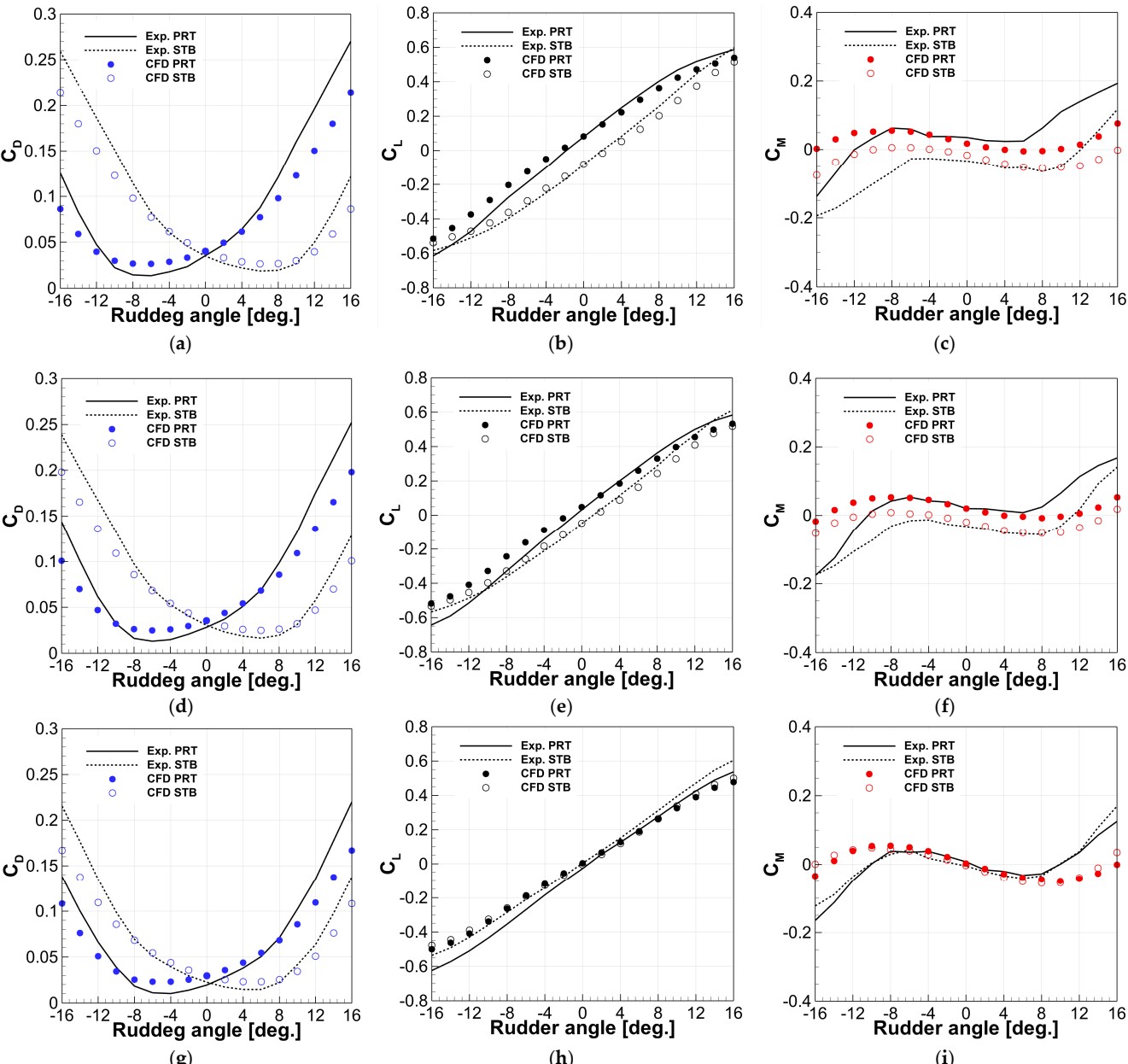

**Figure 19.** Measured and numerically predicted rudder forces and moment for the twisted rudders on the starboard and port side: (**a**) drag force for TR-ini; (**b**) lift force for TR-ini; (**c**) moment for TR-ini; (**d**) drag force for TR-int; (**e**) lift force for TR-int; (**f**) moment for TR-int; (**g**) drag force for TR-fin; (**h**) lift force for TR-fin; (**i**) moment for TR-fin.

### 4.4. Summary of Twisted Rudder Performances

One of the important design criteria in the current work in the development of a twisted rudder in this study was an improvement in the cavitation performance compared with that of the reference flat rudder at the reference rudder angle of $\delta_R = 15°$. At the same reference rudder angle, the maintenance of rudder lift performance which affects the ship's maneuverability, and the reduction of rudder moment applied to the rudder axis were the next important criteria. With respect to the ship's self-propulsion performance, the final thing was to reduce the drag increase of a twin twisted rudder at $\delta_R = 0°$ up to an appropriate level.

Figure 20 shows the hydrodynamic performances of the final designed twisted rudder, TR-fin, related to the cavitation occurrence, forces, and moment. In the numerical analysis, the amount of cavitation was obtained through integration of the cavitation volume, and in

the experiment, it was calculated through image analysis taken from the side and top views of rudder cavitation. The experimental image analysis was done only for the flat and TR-fin rudders. In general, the numerical results were somewhat underestimated compared to the experimental results in each item, but it is seen that the former reflects the experimental trend very well. First, as shown in Figure 20a, the total cavitation amount on the twin TR-fin rudder at $\delta_R = -15°$ was reduced by about 43% and 34.4% in the experiment and numerical prediction, respectively. This is a significant reduction and shows the importance of the accurate prediction of the inflow angles. The computed total cavitation reductions for the twin TR-ini and TR-int rudders were 2.8% and 8.2%, respectively. These values are relatively small, and the reasons can be found in the discussions of Figures 16–18.

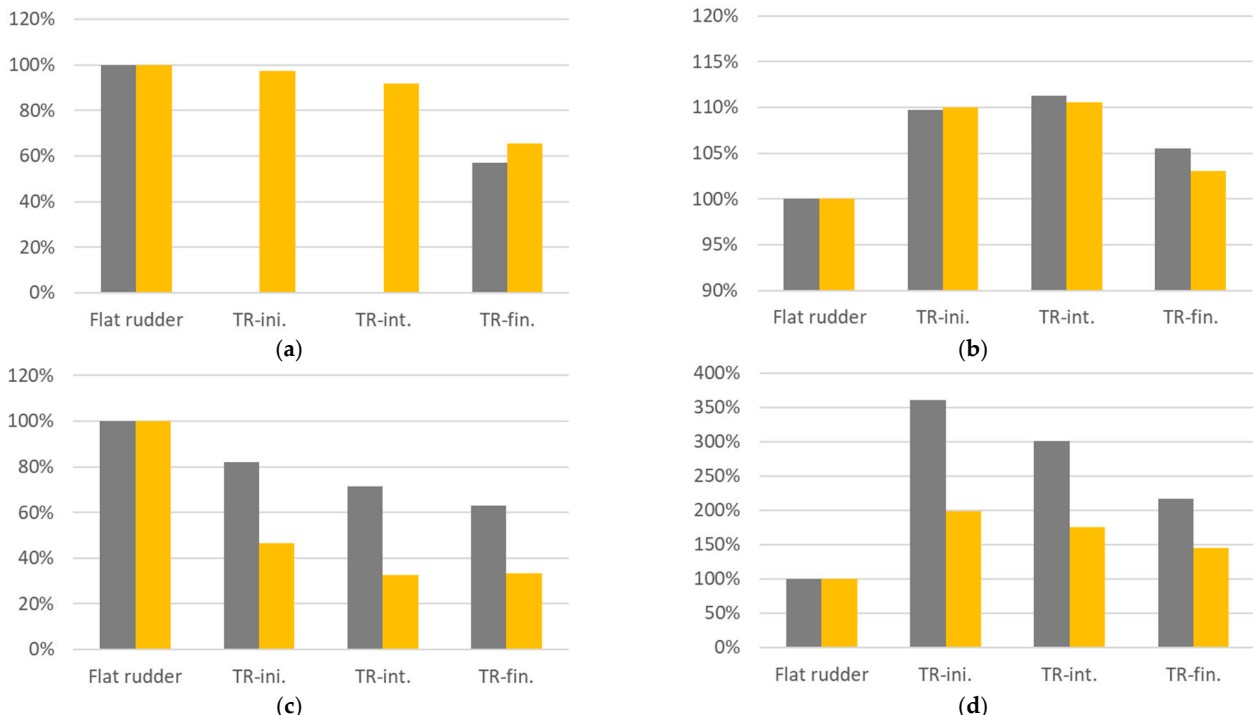

**Figure 20.** Qualitative comparisons of cavitation occurrence, rudder forces, and moment for the twin flat and twisted rudders: (**a**) cavitation amount; (**b**) total lift force; (**c**) maximum moment; (**d**) total drag force.

The predicted total lift forces for each twin twisted rudder at $\delta_R = -15°$ are seen in Figure 20b with that of the twin flat rudder. As discussed in Figure 19, the total lift coefficients for each twin twisted rudder were slightly larger than that of the twin flat rudder by about 6–11% in the measurements and by about 3–11% in the numerical simulations. The measured and predicted total life coefficients were 1.0887 and 0.9429 for the twin TR-fin rudder, respectively, while those values for the twin flat rudder were 1.0319 and 0.9150, respectively. Finally, the twin TR-fin rudder gained about 5.5% and 3% increase in the total lift coefficient in the experiment and numerical simulation, respectively. This implies that the use of twisted rudders will be more effective without a loss of the ship's maneuverability.

The maximum moment coefficients for all the twisted rudders are smaller than that of the flat rudder as seen in Figure 20c. The change in the rudder moment according to twist angle distribution was not critical. However, the maximum moment coefficient was the smallest for the TR-fin rudder. As shown in Figure 20c, the decrease in the rudder moment for the TR-fin rudder was 37% and 66.5% in the experiment and numerical simulation, respectively.

The total drag forces for all the twin twisted rudders at the rudder angle of $\delta_R = 0°$ are relatively larger than that of the twin flat rudder as seen in Figure 20d. It is seen that

the increase in the total drag force is the smallest for the TR-fin rudder. The percentage of the drag increase for the TR-fin rudder was 117% and 45% in the experiment and numerical simulation, respectively. This indicates that the loss of the ship's self-propulsion performance will be minimized by using an optimized twisted rudder. However, other than the method using twist angle distribution, a new design that can reduce the rudder drag force of a twisted rudder below the level of the current flat rudder is necessary.

## 5. Conclusions

The present paper reported a design approach to develop a twisted rudder to improve the rudder performance of an existing surface ship. An accurate twist angle distribution was determined by introducing prediction and correction steps for the inflow angles induced by the propeller. The performances of three twisted rudders with different twist angle distributions were evaluated through both experimental tests in a large cavitation tunnel and CFD simulations. The hydrodynamic characteristics of the twisted rudders were discussed in terms of the total cavitation amount, drag and lift forces, and moment for each twin twisted rudder by comparing with those of a reference twin flat rudder. Due to the symmetry of the twin rudder, the compensation of the induced inflow angles may be unfavorable in the starboard rudder or port rudder for a given rudder angle; such a situation was considered in the current rudder design for a twin twisted rudder.

The total amount of surface cavitation on the final optimized twisted rudders on the starboard and port side decreased largely at the reference design rudder angle of $\delta_R = -15°$ by about 43% and 34.4% in the experiment and numerical prediction, respectively. This shows the importance of compensation of the effect of twist angles and the validity of the present correction method for the inflow angle distribution.

The total lift forces for each twin twisted rudder at the reference design rudder angle were slightly larger than that of the twin flat rudder by about 6–11% found through experimental measurements and by about 3–11% obtained via numerical simulations. Finally, in the final designed twin twisted rudder, there were about 5.5% and 3% increase in the total lift force in the experiment and numerical simulation, respectively. This indicates that the use of twisted rudders is more effective without any loss of the ship's maneuvering performance.

The maximum moment coefficients of all the twisted rudders were smaller than that of the flat rudder. The decrease in the maximum rudder moment for the final designed twisted rudder was 37% and 66.5% as obtained from the experiment and numerical simulation, respectively.

The total drag force for each twin twisted rudders at a rudder angle, $\delta_R = 0°$, increased compared with that of the twin flat rudder. However, the increase in the final designed twin twisted rudder was the least. In addition, the total drag force at rudder angles higher than 4° and 6° in the experiment and numerical simulation, respectively, was smaller than that of the flat rudder, but those of the other twisted rudders were not. This shows that loss in the ship's self-propulsion performance can be minimized by using an optimized twisted rudder. However, the self-propulsion performance should be tested in a towing tank. By using a twisted rudder, rotational energy can be recovered, which in turn, should lead to improvement in the propulsion efficiency.

**Author Contributions:** Conceptualization, I.P. and B.P.; methodology, I.P., J.A., and J.K.; software, J.K.; validation, J.A. and I.P.; investigation, I.P. and J.A.; writing—original draft preparation, I.P.; writing—review and editing, I.P. and J.K.; supervision, B.P.; project administration, B.P. All authors have read and agreed to the published version of the manuscript.

**Funding:** This research was funded by the Ministry of Trade, Industry, and Energy and Defense Acquisition Program Administration and the Agency for Defense Development.

**Acknowledgments:** This study was supported by grants from the dual-use technology project, ''Three-dimensionally curved twisted-rudder technology'' of KRISO (PNS3450).

**Conflicts of Interest:** The authors declare no conflict of interest. The funders had no role in the design of the study; in the collection, analyses or interpretation of data; in the writing of the manuscript, or in the decision to publish the results.

## Nomenclature

| | |
|---|---|
| $c$ | rudder chord length |
| $C_D$ | drag coefficient |
| $C_L$ | lift coefficient |
| $C_M$ | moment coefficient |
| $C_p$ | pressure coefficient |
| $D_P$ | propeller diameter |
| $\mathcal{D}$ | rudder drag force |
| $k$ | turbulent kinetic energy |
| $K_T$ | propeller thrust coefficient |
| $\ell_c$ | rudder moment arm |
| $\mathcal{L}$ | rudder lift force |
| $\dot{m}$ | mass transfer rate |
| $\mathcal{M}$ | rudder moment |
| $n$ | propeller rotational speed |
| $n_o$ | nuclei concentration |
| $N$ | number of cavitation bubbles |
| $p$ | rudder surface pressure |
| $p_o$ | free stream reference pressure |
| $p_T$ | static pressure in the LCT |
| $p_v$ | water vapor pressure |
| $P$ | time averaged pressure |
| $R$ | cavitation bubble radius |
| $R_P$ | propeller radius |
| $s$ | rudder span length |
| $S$ | lateral projected rudder area |
| $t$ | physical time |
| $T$ | propeller thrust |
| $\acute{u}_i$ | velocity fluctuations |
| $U_i$ | time averaged velocities |
| $U_o$ | inflow velocity in the LCT test section |
| $V_s$ | ship speed |
| $\alpha$ | fluid volume fraction |
| $\alpha_i$ | inflow angle |
| $\delta_R$ | rudder angle |
| $\varepsilon$ | turbulence dissipation rate |
| $\sigma_n$ | cavitation number |
| $\mu$ | fluid dynamic viscosity |
| $\mu_t$ | turbulent eddy viscosity |
| $\mu_w$ | water dynamic viscosity |
| $\rho$ | fluid density |
| $\rho_w$ | water density |

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
