# Peer review of "The Prediction of the Performance of a Twisted Rudder"

_applsci, doi:10.3390/app11157098_

Round 1

Reviewer 1 Report

The paper study the performance of a twisted rudder for high-speed boats. The paper presents a comparison study between experimental tests and numerical computation. The paper is easy to follow, the methods, the experimental equipment are well described, as well as the results. The authors can find some comments: 1. The abstract needs more improvement. 2. The authors can check the literature review as some refs and sequence of the paper seem not related to the current paper. 3. The authors can clearly mention the difference between their novel design and the ones used in the literature review. 4. the last sentence, "In near future, self-propulsion and maneuvering tests will be performed including a new idea that can reduce the rudder drag force of twisted rudders below the level of the current flat rudder.", this sentence is not written in a professional way, the author can either mention the new idea or rewrite the whole the sentence. 5. These two sentences can be rewritten to avoid duplication: A) The scale effects on propulsion efficiency for rudder bulb and rudder 60 thrust fin were analyzed based on a numerical method by Shen et al. [24]. Liu et al. [ B) Shin et al. [36] proposed a wavy twisted rudder and experi- 100 mentally and numerically verified the increase of the stalling angle in comparison to a 101 conventional full spade twisted rudder.

Author Response

Point 1: The abstract needs more improvement.

Response 1: The abstract has been updated based on the specific data of the results of our study. You can check it in the attached revised article.

Point 2: The authors can check the literature review as some refs and sequence of the paper seem not related to the current paper.

Response 2: The reviewer's point is correct. [10] and [13] references in the submitted original article lacking in relation to the present work were excluded. Then, the order of references was properly readjusted as noted by the reviewer, and the introduction was revised reflecting the modified order of references.

Point 3: The authors can clearly mention the difference between their novel design and the ones used in the literature review.

Response 3: The differences between the previous studies shown in the literature review and the current study were mentioned on page 3 of the revised article as follows:

“In general, most researches on twisted rudders have been carried out for single screw ships. However, this study focused on a twin twisted rudder of a surface combatant and investigated its effects on hydrodynamic performances”.

Point 4: the last sentence, "In near future, self-propulsion and maneuvering tests will be performed including a new idea that can reduce the rudder drag force of twisted rudders below the level of the current flat rudder.", this sentence is not written in a professional way, the author can either mention the new idea or rewrite the whole the sentence.

Response 4: As pointed out by the reviewer, the last comment in the conclusion seems not to be professional and is judged to be inappropriate expression in the flow of the paper. So, we have excluded this sentence from the conclusions

Point 5: These two sentences can be rewritten to avoid duplication: A) The scale effects on propulsion efficiency for rudder bulb and rudder thrust fin were analyzed based on a numerical method by Shen et al. [24]. Liu et al. [ B) Shin et al. [36] proposed a wavy twisted rudder and experimentally and numerically verified the increase of the stalling angle in comparison to a 101 conventional full spade twisted rudder.

Response 5: Following the reviewer' comments, the sentences about the researches conducted by Shen et al [24] and Shin et al [36] mentioned in the introduction of the pre-revision paper was revised as follows:

“Shen et al. [18] investigated the scale effects on ship with ESDs using CFD simulations and reported the reliability of CFD data at full scale compared to model scale”.

*Additional comments for the reviewer: Although there was no mention made by the reviewer, minor revisions were done by the authors to improve the paper, and each revision can be found through track changes in the revised paper. And in the case of Figure 20, the result of the rudder angle of -15° should be entered, but the result of -16° was used. So I corrected it with the result for the rudder angle of -15°.

Reviewer 2 Report

1) Abstract – in the Abstract should be added some of the most important obtained results (its exact values). Without it, the Abstract seems to be only descriptive and general.

2) Due to many abbreviations, symbols and markings used throughout the paper, it will be very helpful to any reader that the Authors add in the paper a Nomenclature inside which all of them will be presented and explained in one place. Without a Nomenclature, it is sometimes hard to track a way of thinking in the paper (the reader must constantly turn back throughout the paper to properly understand all the presented elements). Adding a Nomenclature will completely resolve mentioned problem.

3) Line 208 – this is Equation 6, not Equation 5.

4) In the Subsection 3.3. should be presented at least general specifications of the computer used for the performing simulations. According to presented computational domain, it seems that these simulations require usage of the supercomputer – therefore, a presentation of computer specifications is required.

5) For the results obtained by measurements, the Authors should present at least general specifications and type of each used measuring device. How measuring equipment accuracy and precision influences obtained and presented measurement results?

6) In the paper should be added a description and definition of all three coefficients presented in Figure 6.

Final remarks: This is very interesting and novel research with clear and obvious scientific novelty. The above mentioned corrections/additions should be performed with an aim that all the elements of performed research become completely clear. After performing above mentioned corrections/additions, this paper will have my full recommendation for publication.

Author Response

Point 1: Abstract – in the Abstract should be added some of the most important obtained results (its exact values). Without it, the Abstract seems to be only descriptive and general.

Response 1: The abstract has been updated based on the specific data of the results of our study. You can check it in the attached revised article.

Point 2: Due to many abbreviations, symbols and markings used throughout the paper, it will be very helpful to any reader that the Authors add in the paper a Nomenclature inside which all of them will be presented and explained in one place. Without a Nomenclature, it is sometimes hard to track a way of thinking in the paper (the reader must constantly turn back throughout the paper to properly understand all the presented elements). Adding a Nomenclature will completely resolve mentioned problem.

Response 2: As noted by the reviewer, a nomenclature was prepared and provided at the end of the introduction in the revised paper. At the same time, while organizing the variables, the names of some variables used in the paper were modified.

Point 3: Line 208 – this is Equation 6, not Equation 5.

Response 3: While adding the definitions of the drag, lift, and moment coefficients along with the pressure coefficient suggested by the reviewer in the 6th comment, the wrong equation number was also corrected.

Point 4 In the Subsection 3.3. should be presented at least general specifications of the computer used for the performing simulations. According to presented computational domain, it seems that these simulations require usage of the supercomputer – therefore, a presentation of computer specifications is required.

Response 4: We used a Linux based HPC (high performance computing) system for computations. A brief specification about the system was presented in the revised article as follows:

“The present computations were carried out on a Linux-based PC cluster system whose each node has 20 Intel®_Xeon® 2.40 GHz processors. All runs used a total of 280 processors.”

Point 5: For the results obtained by measurements, the Authors should present at least general specifications and type of each used measuring device. How measuring equipment accuracy and precision influences obtained and presented measurement results?

Response 5: General specifications and type of each used measuring device were provided in the revised paper with their uncertainties as follows:

“Fourteen pressure taps were installed along the rudder surface at 60% span of the rudder. The relative pressure was measured with the pressure transducers made by Validyne DP15. The pressure was measured with the uncertainty of 0.18%”.

“A dynamometer was installed in the stern hull over the rudder to measure the drag force () in the x direction, the lift force () in the y direction, and the moment () in the z direction acting on the rudder. The dynamometer was manufactured by Wonbang Forcetech Co. Ltd. and can measure drag, lift, and moment up to 1000 N, 2000 N, and 45 Nm, respectively with uncertainties of 0.38%, 0.23%, and 1.70%”.

Point 6: In the paper should be added a description and definition of all three coefficients presented in Figure 6.

Final remarks: This is very interesting and novel research with clear and obvious scientific novelty. The above mentioned corrections/additions should be performed with an aim that all the elements of performed research become completely clear. After performing above mentioned corrections/additions, this paper will have my full recommendation for publication.

Response 6: We have provided the definitions of the drag, lift, and moment coefficients along with the pressure coefficient in equations (1), (4), (5), and (6), respectively.

*Additional comments for the reviewer: Although there was no mention made by the reviewer, minor revisions were done by the authors to improve the paper, and each revision can be found through track changes in the revised paper. And in the case of Figure 20, the result of the rudder angle of -15° should be entered, but the result of -16° was used. So I corrected it with the result for the rudder angle of -15°.
